# The Influence of Heat and Cryogenic Treatment on Microstructure Evolution and Mechanical Properties of Laser-Welded AZ31B

**DOI:** 10.3390/ma16134764

**Published:** 2023-06-30

**Authors:** Yulang Xu, Peng Qian, Yanxin Qiao, Wujia Yin, Zhiwei Jiang, Jingyong Li

**Affiliations:** Advanced Welding Technology Provincial Key Laboratory, Jiangsu University of Science and Technology, Zhenjiang 212003, China; yulang_xu@163.com (Y.X.);

**Keywords:** AZ31B, laser welding, solution treatment, cryogenic treatment

## Abstract

The pores and coarse lamellar Mg_17_Al_12_ that inevitably occur in the weld zone are the major challenge for laser-welded magnesium (Mg) alloys including AZ31B. In order to improve microstructure uniformity and eliminate welding defects, a new process assisted with combination of heat and cryogenic treatment was applied in this study. The results showed that after solution treatment, the number and size of precipitates decreased and the uniformity of the microstructure improved. After cryogenic treatment, the lamellar Mg_17_Al_12_ was cracked into particles, and the grain size was refined. After solution + cryogenic treatment, Al_8_Mn_5_ substituted the lamellar Mg_17_Al_12_. Through studying the changes in microhardness, precipitates, and microstructure under different treatments, it was found that the conversation of Mg_17_Al_12_ from lamellar state into particle-like state as well as the appearance of dispersed Al_8_Mn_5_ particles played a second-phase strengthening role in improving the mechanical properties of Mg alloy laser-welded joint, and the tensile strength (258.60 MPa) and elongation (10.90%) of the sample were 4.4% and 32.6% higher than those of the as-welded joint.

## 1. Introduction

In September 2020, China proposed the “carbon neutrality” and “emission peak” strategies which have attracted worldwide attention. Extensive application of the lightest structural materials, magnesium alloys, is one of the best strategies to achieve the goal for weight-saving and CO_2_ emission reduction. In addition, Mg alloys have high specific strength and stiffness, superior damping performance, good biocompatibility, and other advantages. Hence, magnesium and its alloys have been applied in aerospace, automobile, automotive, electronics industries, and other manufacturing areas [1,2,3,4,5]. Practical applications include car seats, wheel hubs, aircraft skin frame air control surfaces, laptop casings, etc. [6,7,8,9,10].

Among the modern welding techniques of Mg alloys, laser welding has been considered to be of great potential because of high welding speed, precise control of power output, narrow joints with reduced heat affected zone (HAZ), low distortion, and excellent environment adaptability [11,12,13]. Even though laser welding is a convenient method for Mg alloy welding, there still exist a series of defects in the welded joint, such as cracks, pores, lamellar Mg_17_Al_12_ phase, etc. [14,15,16]. Based on the issues and challenges identified above, some future research directions are suggested. In recent years, there have been many studies on the improvement of the abovementioned defects by heat treatment technology [17,18,19,20], while there have been few studies on cryogenic treatment technology; especially little attention is paid to laser-welded joints [21,22,23,24]. Shen et al. [25] found that most of dendric Mg_17_Al_12_ was choppy under cryogenic treatment, which deteriorated the ductility of the welded joint. Liu et al. [26] proved that cryogenic treatment could be efficient only for the formation of a precipitate with a much lower atom density than the matrix. Chen et al. [27] demonstrated that a large amount of thermal stress and deformation energy were generated inside the welded joint after cryogenic treatment, leading to grain refinement and homogenization, and its effect was negatively correlated with cryogenic temperature.

In this paper, the novelty of this work consists in a new process of laser welding of the Mg alloy AZ31B assisted with a combination of heat and cryogenic treatment which was applied to improve microstructure uniformity and enhance mechanical properties so as to broaden the future application of Mg alloys.

## 2. Experimental Section

### 2.1. Materials and Welding Equipment

The material used in this welding experiment was commercial rolled magnesium alloy AZ31B (Al 3.19 wt.%, Zn 1.25 wt.%, Mn 0.34 wt.%, Si 0.12 wt.%, Mg Bal.) with a thickness of 4 mm. An IPG YLS-6000-S2-TR fiber laser with a wavelength of 1070 nm was applied in the experiment. The laser beam was transmitted through a processing fiber and focused on the workpiece surface by a focusing lens with a 310 mm focal length. Accordingly, the spot size of a focused laser beam is approximately 0.29 mm. High-purity argon (99.99%) was supplied coaxially as shielding gas to avoid the oxidation of melt pool. The laser welding parameters were adopted as follows: laser power, 2.5 kW; welding speed, 30 mm/s; focal position, 0 mm; shielding gas flow rate, 20 L/min.

### 2.2. Post-Weld Treatment System

The AZ31B plate was first welded through laser welding and then processed into standard samples by wire cutting. The samples were separated into four groups lambed as A, B, C and D, as shown in Table 1. The sample of group A was the as-welded joint. The sample of group B was only solution treated at 410 °C for 6 h. The sample of group C was put into liquid nitrogen directly for 12 h and then transferred into the atmosphere. The sample of group D was treated with solution in advance and then put into liquid nitrogen for 12 h.

### 2.3. Testing Equipment

The microstructure of the welded joint was observed by optical microscope (SZ61 and BX51M; Olympus Co., Ltd., Tokyo, Japan). Element content and distribution pattern in the weld joint were examined by energy dispersive spectrometer (S3400N; Hitachi Co., Ltd., Tokyo, Japan). Phase composition of the weld zone was tested by X-ray diffractometer (SmartLab9kW; Rigaku Co., Ltd., Tokyo, Japan) with a scanning step of 0.01° and a scanning speed of 10°/min. Microhardness measurements were performed on the transverse cross-section of the welded joints using a Vickers hardness tester (KB30S, KB Co., Ltd., Assenheim, Germany), and the measurements were obtained at intervals of 0.2 mm under a load of 100 g for 15 s. Tensile tests were performed with a tensile rate of 2 mm/min at room temperature using a universal testing machine (CMT5205, MTS, Minneapolis, MN, USA). Fracture morphologies were obtained from a scanning electron microscope (S3400N; Hitachi Co., Ltd., Tokyo, Japan).

## 3. Results and Discussion

### 3.1. Microstructure Evolution Analysis

Figure 1a,a′,a″ present the microstructure of the as-welded joint. Figure 1a shows the typical microstructure near the fusion boundary in the welded joint, from left to right. The welded joints can be divided into base metal (BM), heat-affected zone (HAZ), fusion line (FL) and weld zone (WC). As seen in Figure 1a′, typical equiaxed grain presented in the weld center and a large number of black precipitated particles dispersed in the grains and grain boundaries. It can be clearly seen that the grain boundary is blurred due to the precipitation of a large number of segregations by non-equilibrium solidification; it is generally believed that the black precipitated particles are segregations of supersonic metal atoms for the rapid cooling speed of laser beam welding [28]. The grain size in the weld center is close to 10 μm, which is smaller than that in the BM (20~25 μm), indicating that grain refinement occurs after laser welding.

Figure 2a,a′,a″ present the microstructure of the welded joints under different treatments. As seen from Figure 2a′, after solution treatment, the average grain diameter in the weld center grew significantly, nearly reaching 35 μm. The reasons may be as follows: on the one hand, the solution treatment temperature (410 °C) is higher than the recrystallization temperature of 0.4 Tm (melt temperature) [29]; it can be inferred that the grain continued to grow, absorbing the remaining energy after static recrystallization; on the other hand, residual stress can provide the driving force for recrystallization and growth, too. In addition, the coarse black particles in the welded joint diminished remarkably, and only a small amount of black dot-like particles remained.

As shown in Figure 2b,b′,b″, after cryogenic treatment, the precipitation of the second phase particles mainly occurs due to the volume shrinkage of the sample as it rapidly cools from room temperature to liquid nitrogen temperature and generates large compressive stress and deformation energy inside the welded joint. The deformation energy can be used to provide the driving force for second-phase precipitation. In addition, the volume contraction and the decrease in lattice constant caused by ordered solid solution also leads to the precipitation of the second phase. Generally speaking, magnesium alloys typically undergo twinning during deformation, and although the formation of twin crystal during cryogenic process has been reported in Jiang et al.’s research, the mechanism behind this phenomenon is currently unclear.

As shown in Figure 2c,c′,c″, compared with the solution treated sample B, when the solution-treated laser-welded joint was followed by a cryogenic-treated one, both the number and size of precipitates were significantly reduced, and the grain size was refined. Compared with the only cryogenic-treated sample C, the microstructure of sample D became more uniform and the distribution of precipitates in the welded joint was more dispersed.

### 3.2. Phase Composition Analysis

#### 3.2.1. XRD Analysis

As shown in Figure 3, the category of phases in the XRD diffraction patterns of the as-welded, solution-treated, cryogenic-treated and solution + cryogenic-treated joints have not changed, and all of them were composited by α-Mg and β-Mg_17_Al_12_. Meanwhile, this has little effect on the diffraction peak intensity of the β-Mg_17_Al_12_ phase, because the second phase was composed of a small amount of Mg (Al, Zn) compounds and eutectic β-Mg_17_Al_12_ which is difficult to scan.

However, after the solution + cryogenic treatment, the grains underwent varying degrees of rotation and orientation changes, resulting in preferred grain orientation. The relative strengths of the three strong peaks, from weak to strong, ($10\bar{1}0$), (0002), ($10\bar{1}1$), are closer to the standard card of Mg. Table 2 shows the lattice constants of welded joints after composite treatment. It can be observed that severe lattice distortion occurred after cryogenic treatment. However, if the welded joint were treated with solution in advance, the degree of lattice distortion would decrease, presenting a smaller axial ratio and higher lattice symmetry.

#### 3.2.2. SEM Analysis

Figure 4 shows the SEM morphologies in the weld zone of the welded joint treated under different treatments, and the chemical composition of testing points are shown in Table 3. As seen from Figure 4a, plenty of white spherical particles were distributed in the α-Mg matrix, and it was demonstrated that they may be the products of a eutectic Mg_17_Al_12_ phase in the magnesium alloy [30]. This precipitated phase is suggested to usually occur in the areas that are rich in aluminum [31]. It is obviously shown in Table 3 that the content of Al in the white spherical particle A2 (7.24%) was twice higher than that in the α-Mg matrix A1 (3.65%).

As shown in Figure 4b, the white coarse particles diminished remarkably after solution treatment, and only a small amount of fine white particles existed at the grain boundary. The EDS analysis results in Table 3 showed that the percentage content of aluminum element in the α-matrix B1 increased to 4.62%, which is obviously higher than that in the as-welded joint, indicating that the precipitated atoms dissolved into the α-matrix partially.

In the next cryogenic process, it can be clearly seen from Figure 4c that compared with the as-welded joint, the coarse precipitates were mostly cracked into many pieces because of the difference of the coefficient of thermal expansion and contraction. In the testing point C2, residual lamellar precipitates can be seen after cryogenic treatment. The content of Al reached 13.43%, which is obviously higher than that in testing point C1, while the content of Mg was 85.65%. In theory, atoms are prone to transition to defects during the diffusion process, leading to the formation and aggregation of the second phase at the defects.

As shown in Figure 4d, it can be inferred that many aluminum atoms dissolved within basic α-Mg after solution treatment, and then the cracked Mg17Al12 pieces narrowed down into particle-like precipitates under the function of cryogenic treatment. Comparing the element content test results of testing points D1 (1.90%, 0.20%) and D2 (5.31%, 4.36%), it was found that the content of the aluminum element and manganese elements in D2 was significantly higher than that of the α-matrix. According to the Al-Mn binary phase diagram and the molar ratio (nearly 8:5) of aluminum to manganese in the white particle, the which are identified as Al_8_Mn_5_. In the cryogenic process, a great quantity of manganese was separated while forming Al_8_Mn_5_ that consumed a lot of aluminum atoms; it is probably responsible for the decrease in both the number and volume of Mg_17_Al_12_.

### 3.3. Microhardness Analysis

Figure 5 shows the microhardness distribution of the laser-welded joint of magnesium alloy AZ31B treated under different treatments. All samples underwent three measurements. For sample A, the microhardness in the heat-affected zone (HAZ) is not lower than that in the weld zone (WZ), and it may be caused by Mg_17_Al_12_. It can be observed that after solution treatment, the microhardness of sample B in the weld zone (WZ) decreased compared with that in the as-welded joint.

In comparison with samples A and B, the microhardness of sample C improved obviously after cryogenic treatment, especially in the base metal (BM). This is because during the cryogenic treatment process, the welded joint was subjected to compressive stress, resulting in grain shrinkage and refinement. The increased grain boundary area hinders the movement of dislocations, leading to an increase in the resistance of the micro zone to plastic deformation and an increase in microhardness. In addition, the dispersed distribution of the broken second phase has a higher hardness, which plays the second-phase strengthening role and finally leads to an increase in the microhardness of the entire joint. As shown, the microhardness of sample D improved significantly compared with samples A and B; it can be inferred that cryogenic treatment played a more important role than solution treatment for the improvement of microhardness, and the main reason for the improvement of microhardness could be ascribed to the generation of Al_8_Mn_5_ which substituted the coarse Mg_17_Al_12_.

### 3.4. Tensile Properties

#### 3.4.1. Comparison of Tensile Properties

Figure 6 shows the stress–strain curves [32] and tensile properties of the as-welded joints and those subjected to different treatments. All samples underwent three measurements. As shown in Figure 6b, after solution treatment, the tensile strength and elongation improved from 247.69 MPa, 8.22% to 253.15 MPa, 11.91%, respectively, for the homogeneous microstructure compared with the as-welded joint.

After cryogenic treatment, there appeared many scattered and fragmented particles in the welded zone, which may be responsible for its better mechanical properties, meanwhile the toughness declined slightly compared with sample B. The cryogenic treatment emerged from the above considerations as an effective method to improve the static mechanical properties of the AZ31B magnesium alloy.

It can be observed that sample D has not only a high tensile strength of 258.60 MPa that is 4.4% higher than that of sample A (247.69 MPa), but also a better toughness than sample C. It may be the appearance of smaller dispersed Al8Mn5 and the conversion of lamellar Mg17Al12 into particles that is responsible for sample D displaying good comprehensive mechanical properties.

#### 3.4.2. Comparative Analysis of Fracture Surfaces

Figure 7a–d,a′–d′ show the fracture morphologies of the tensile samples that underwent different treatments, and a′, b′, c′, d′ is the corresponding local enlarged image of a, b, c, d. The diagnosis of fracture properties is a prerequisite for the cause of fracture formation, while the fracture mechanism is the deepening and extension of the cause of fracture formation.

As seen in Figure 7a, the fracture of the as-welded joint appears dark gray, belonging to intergranular fracture. The reason is that there is a large amount of component segregation on the grain boundary, which leads to the aggregation of precipitates, then resulting in lower grain boundary bonding force than intergranular bonding force and crack propagation along the grain boundary. In Figure 7a′, there is a cleavage step shown in Area Ⅰ and some dimples shown in Area Ⅱ in the fracture surface, which presents a typical quasi-cleavage fracture [33]. This is due to the fact that α-Mg has a close-packed hexagonal crystal structure with a less slip system and is not easily to plastic deformation. In addition, the continuous, hard and brittle β-Mg_17_Al_12_ reinforcement phase is distributed in the α-Mg matrix, which hinders the movement of dislocations in the α-Mg matrix during plastic deformation and induces stress concentration at the interface between reinforcement phase and matrix phase which acts as the source of microcracks, eventually leading to fracture.

Figure 7b presents the fracture morphologies after solution treatment. The shiny fracture surface indicates that it belongs to a trans-granular fracture. In addition, in Figure 7b′, it can be seen that the cleavage platform almost disappeared, a number of dimples with different sizes are distributed unevenly on the fracture surface, and the number of dimples is significantly higher than that of the as-welded joints, exhibiting quasi-cleavage cracks. These observations are in agreement with high elongation (10.90%) of the welded joint.

Figure 7c presents the fracture morphologies after cryogenic treatment. The shiny fracture surface indicates that it belongs to a trans-granular fracture. In Figure 7c′, it can be seen that there are obvious river patterns, cleavage surfaces and steps; only a small number of dimples exist compared with the solution-treated sample, exhibiting more brittle crack characters belonging to cleavage fracture. The reason for poor toughness is that the lattice distortion caused by cryogenic treatment is more severe, and the stress concentration is more obvious. In addition, the disorderly distribution of the broken Al_17_Mg_12_ phase hinders the movement of dislocations during plastic deformation, inducing stress concentration at the interface between reinforcement phase and matrix phase, which acts as the source of microcracks and eventually leading to fracture.

Figure 7d presents the fracture morphologies after solution + cryogenic treatment. The shiny fracture surface indicates that it belongs to trans-granular fracture. In Figure 7d′, the dimples with different sizes exhibit the characteristics of ductile fracture, which belongs to quasi-cleavage fracture, while cleavage planes can still be observed. Unlike cleavage fracture, the crack source of the quasi-cleavage fracture is generated by voids and the second phase inside the grain. This is because the dispersed fine particle Al_8_Mn_5_ affects the crack propagation, and it is difficult to strictly follow a certain crystallography plane when the crack propagates in the grain. On the contrary, the distribution of small particles Al_8_Mn_5_ plays a decisive role in the propagation path of the cracks. The fracture occurred in the area where fewer particles of Mg_17_Al_12_ and Al_8_Mn_5_ were precipitated. It may suggest that diffuse-distributed Al_8_Mn_5_ can be responsible for the improved tensile strength.

## 4. Conclusions

(1)Solution treatment played the role of regulating the organization, reducing segregation, and improving the tensile strength of magnesium alloys. The grain size was refined under the function of cryogenic treatment, and most of lamellar Mg_17_Al_12_ was choppy.(2)After solution + cryogenic treatment, the conversion of Mg_17_Al_12_ from a lamellar state into a particle-like state as well as the appearance of dispersed Al_8_Mn_5_ particles played a major role in improving the mechanical properties of the welded joint. The tensile strength (258.60 MPa) and elongation (10.90%) of the solution + cryogenic-treated sample were 4.4% and 32.6% higher than that of the as-welded joint, and the fracture morphology exhibits quasi-cleavage fracture characteristics.(3)The hardness of the solution-treated sample in the weld zone decreased compared with that in the as-welded joint, while the hardness of the welded joint followed by cryogenic treatment or solution + cryogenic treatment was greatly improved, and was significantly higher than that of the as-welded joint.

## Figures and Tables

**Figure 1 materials-16-04764-f001:**
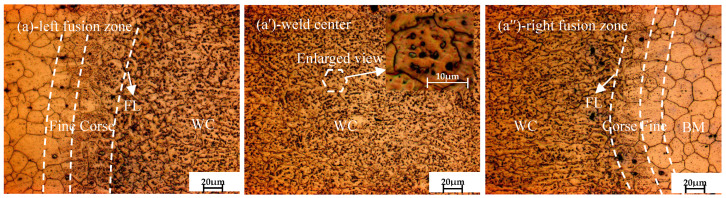
Microstructure observation of the as-welded joint.

**Figure 2 materials-16-04764-f002:**
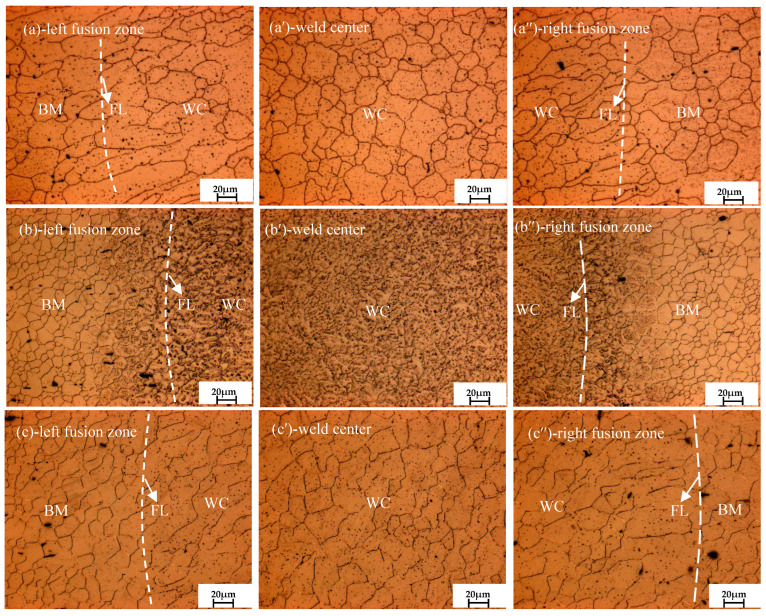
Microstructure observation of welded joints under different treatment.

**Figure 3 materials-16-04764-f003:**
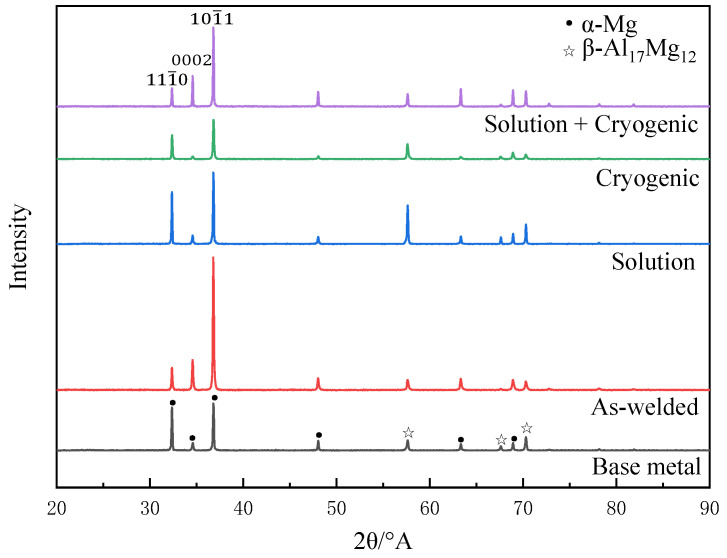
XRD patterns of weld zone under different treatment states.

**Figure 4 materials-16-04764-f004:**
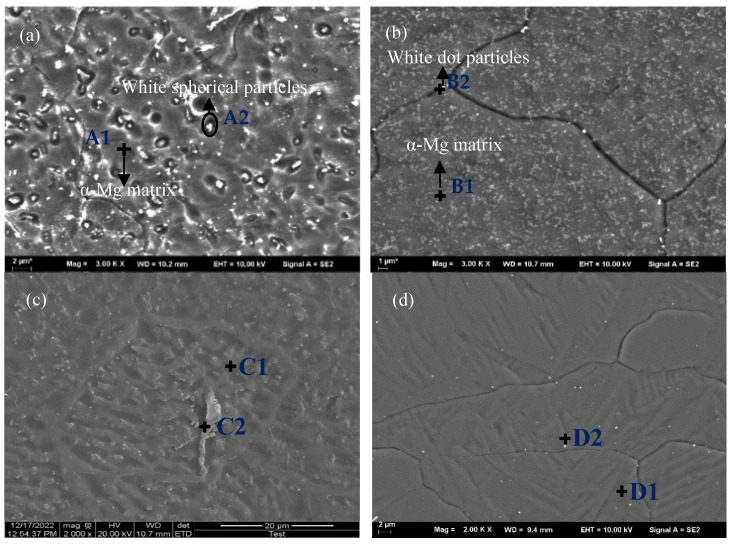
EDS spot scanning of characteristic points. (**a**) As-welded; (**b**) Solution-treated; (**c**) Cryogenic-treated; (**d**) Solution + cryogenic-treated.

**Figure 5 materials-16-04764-f005:**
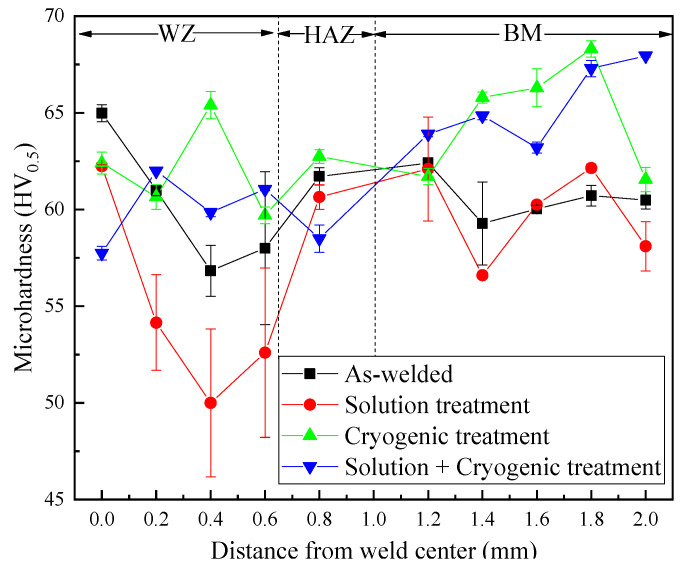
Microhardness distribution of welded joints under different treatments. Sample A: As-welded; Sample B: Solution-treated; Sample C: Cryogenic-treated; Sample D: Solution + cryogenic-treated.

**Figure 6 materials-16-04764-f006:**
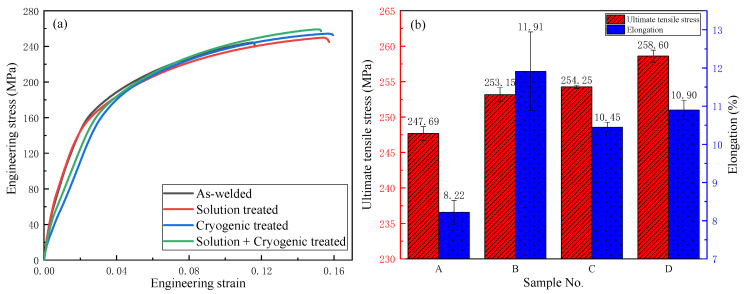
Curves of stress–strain and tensile properties of different treatments. (**a**) Engineering stress–strain; (**b**) Tensile properties.

**Figure 7 materials-16-04764-f007:**
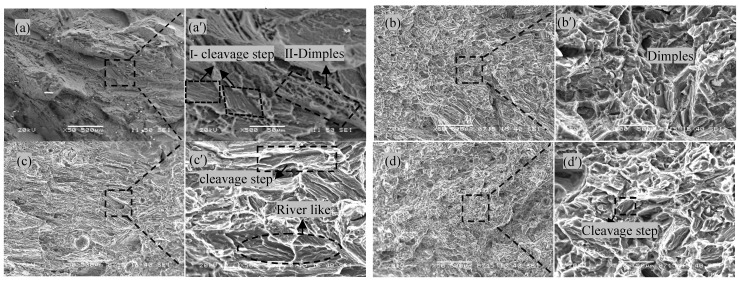
Fractographies of tensile specimens. (**a**) As-welded; (**b**) Solution-treated; (**c**) Cryogenic-treated; (**d**) Solution + cryogenic-treated.

**Table 1 materials-16-04764-t001:** Treatment states.

Sample No.	Solution Treatment	Cryogenic Treatment
A	—	—
B	410 °C—6 h	—
C	—	−196 °C—12 h
D	410 °C—6 h	−196 °C—12 h

**Table 2 materials-16-04764-t002:** Lattice constants of weld zone under different treatment states.

Sample No.	a (Å)	c (Å)	c/a
As-welded	3.19414	5.18870	1.624444
Solution-treated	3.19584	5.19012	1.624024
Cryogenic-treated	3.19547	5.17912	1.620769
Solution + cryogenic-treated	3.19348	5.18851	1.624720

**Table 3 materials-16-04764-t003:** Mass fraction of elements at characteristic points of EDS spot scanning (wt.%).

Sample No.	Characteristic Points	Mg	Al	Zn	Mn
A	A1 (White spherical particles)	95.38	3.65	0.91	0.06
	A2 (α-Mg matrix)	89.13	7.24	2.92	0.71
B	B1 (α-Mg matrix)	94.34	4.62	0.95	0.09
	B2 (White dot particles)	92.17	7.04	0.59	1.20
C	C1 (α-Mg matrix)	95.95	3.16	0.64	0.24
	C2 (Lamellar precipitation)	85.65	13.43	0.48	0.44
D	D1 (α-Mg matrix)	96.85	1.90	1.05	0.20
	D2 (White dot particles)	89.56	5.31	0.77	4.36

## Data Availability

All data generated or analyzed during this study are included in this published article.

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
