# Peer review of "The Influence of Heat and Cryogenic Treatment on Microstructure Evolution and Mechanical Properties of Laser-Welded AZ31B"

_materials, 2023, doi:10.3390/ma16134764_

Round 1
Reviewer 1 Report
1- The novelty of this work and how it differs from earlier research should be addressed by the authors.
2- Nothing is mentioned in the text about the uncertainty in the measurements, so please discuss this issue in the text.
3- Please describe how the cryogenic treatment was carried out in the text.
4- The conclusions section needs to be improved.
Overall the manuscript appears to be clearly and carefully written.
Author Response
We have studied comments carefully and have made correction, which we hope meet with approval.

Reviewer 2 Report
The refereed manuscript is devoted to study the influence of heat and cryogenic treatment on microstructure evolution and mechanical properties of laser-welded AZ31B. The authors state that “it was found that the conversation of Mg17Al12 from lamellar state into particle-like state as well as the appearance of dispersed Al8Mn5 particles played a second phase strengthening role in improving the mechanical properties of Mg alloy laser welded joint, and the tensile strength (259.2MPa) and elongation (10.7%) of the sample were 4.4% and 27.0% higher than that of the as welded joint.”
Some comments on the manuscript are following:
1- The Introduction does not provide sufficient background. The introduction does not explain the major contributions and novelty of this work. The significance of the proposed solution has not been summed up.
2- The discussion of the results suffers from the physics of the problem. The authors essentially limit themselves to describing the captions of the figures in an enlarged way or to illustrate what is observed in them without entering into justifying or giving arguments that support these results.
3- The references are not enough, especially in the “Introduction” section.
4- According to the title of the manuscript, it will be better understanding and interesting for reading the manuscript if some practical application (example) of the study is presented in the manuscript.
5- English should be thoroughly checked.
English should be thoroughly checked.
Author Response

(The authors gave the same response as above.)

Reviewer 3 Report
In this work, the authors have proposed a cryogenic + solution treatment to improve microstructure uniformity and welding defects for laser welding of Mg alloy AZ31B. They have claimed that a combination of cryogenic and solution treatment can decrease the size and number of precipitates and improve the uniformity of the microstructure. Although the manuscript is nicely written, a few aspects need to be clarified. Therefore, the reviewer recommends a major revision. Reviewer has the following comments.
1. Regarding microhardness measurement in Figure 5, how many measurements were done for each condition? The plot should include an error bar for each location to depict whether the hardness increase/decrease is within the error bar.
2. How many tensile tests were done for each condition? The plot in Figure 6b should include an error bar for each condition. It will again depict whether the authors' claim of UTS improvement is within the error bar of each other or not.
3. Instead of using the ‘Tensile property’ in the Y-axis of Figure 6b, the authors should use ultimate tensile stress or UTS.
4. As the authors show that the only solution-treated welds have good ductility (11.18) and reasonable UTS value (253.86 MPa) compared to Cryogenic treated + solution treated, what is the justification for using more expensive Cryogenic treatment? The authors should clearly explain this.
5. In Figure 6a, the authors should modify the X and Y axes with Engineering stress/strain or True stress-strain. The legends for the green line should be changed to ‘Cryogenic + Solution treated.’
6. The elastic moduli presented in the plots of Figure 6a are much low for magnesium. How was the strain measured in those tests? If the crosshead displacement was used to calculate strain, a large amount of system/tooling compliance must be accounted for. Otherwise, the tests will overpredict the actual strain in the tensile specimens. Please see the thesis of Christopher Finfrock to demonstrate the effect, along with code to correct the data properly. https://www.researchgate.net/post/How-to-do-compliance-correction-for-a-given-stress-strain-data
Author Response

(The authors gave the same response as above.)

Round 2
Reviewer 1 Report
Overall the manuscript appears to be clearly and carefully written. The authors have made satisfactory amendments to the manuscript in response to my previous comments and concerns. In my opinion, the manuscript contains now all information and is suitable for publication in materials.
Overall the manuscript appears to be clearly and carefully written.
Reviewer 2 Report
The authors have revised the manuscript according to my report carefully. I recommend the acceptance of this manuscript.
Minor editing of English language required.
Reviewer 3 Report
The manuscript is revised satisfactorily. It can be accepted in its present form.